# Effect of Different Maxillary Oral Appliance Designs on Respiratory Variables during Sleep

**DOI:** 10.3390/ijerph19116714

**Published:** 2022-05-31

**Authors:** Kay Thwe Ye Min Soe, Hiroyuki Ishiyama, Akira Nishiyama, Masahiko Shimada, Shigeru Maeda

**Affiliations:** 1Masticatory Function and Health Science, Graduate School of Medical and Dental Sciences, Tokyo Medical and Dental University, Tokyo 113-8549, Japan; kay.ofpm@tmd.ac.jp (K.T.Y.M.S.); h.ishiyama.rpro@tmd.ac.jp (H.I.); 2Department of General Dentistry, Graduate School of Medical and Dental Sciences, Tokyo Medical and Dental University, Tokyo 113-8549, Japan; 3Dental Anesthesiology and Orofacial Pain Management, Graduate School of Medical and Dental Sciences, Tokyo Medical and Dental University, Tokyo 113-8549, Japan; mshimada.ofpm@tmd.ac.jp (M.S.); maedas.daop@tmd.ac.jp (S.M.)

**Keywords:** maxillary oral appliance, respiratory event index, sleep quality, sleep apnea

## Abstract

This study aimed to analyze the efficacy of maxillary oral appliance (MOA) designs on respiratory variables during sleep. At baseline, 23 participants underwent a sleep test with a portable device for two nights and were categorized as participants with mild obstructive sleep apnea (mild-OSA) (*n* = 13) and without OSA (w/o-OSA) (*n* = 10). Three types of MOAs, standard-OA (S-OA), palatal covering-OA (PC-OA), and vertically increasing-OA (VI-OA), were each worn for three nights, and sleep tests with each MOA were performed with a portable device for two nights. Based on the average of the respiratory event index (REI) values for the two nights for each MOA, w/o-OSA participants with an REI ≥ 5.0 were defined as the exacerbation group and those with an REI < 5.0 as the non-exacerbation group. In mild-OSA participants, an REI ≥ 15.0 or REI ≥ baseline REI × 1.5 were defined as the exacerbation group and those with an REI < 15.0 and REI < baseline REI × 1.5 were defined as the non-exacerbation group. The percentage of the exacerbation and non-exacerbation groups with MOA was evaluated in the w/o-OSA and mild-OSA participants. The maxillary and mandibular dental-arch dimension was compared by dentition model analysis. The exacerbation group in w/o-OSA participants (*n* = 10) comprised 10.0% participants (*n* = 1) with S-OA, 40.0% (*n* = 4) with PC-OA, and 30.0% (*n* = 3) with VI-OA. The exacerbation group in the mild-OSA participants (*n* = 13) comprised 15.4% subjects (*n* = 2) with S-OA, 23.1% (*n* = 3) with PC-OA, and 23.1% (*n* = 3) in VI-OA. In the model analysis for w/o-OSA, the posterior dental arch width was significantly greater in the exacerbation group than in the non-exacerbation group wearing S-OA (*p* < 0.05). In addition, the ratio of the maxillary to mandibular dental arch width (anterior dental arch width) was significantly greater in the exacerbation group than in the non-exacerbation group for both PC-OA and VI-OA (*p* < 0.05). In mild-OSA, the maxillary and mandibular dental arch lengths and the ratio of maxillary to mandibular dental arch width (posterior dental arch width) were significantly smaller in the exacerbation group than in the non-exacerbation group for S-OA (*p* < 0.05). This study confirmed that wearing an MOA by w/o-OSA and mild-OSA participants may increase the REI during sleep and that PC-OA and VI-OA may increase the REI more than S-OA. The maxillary and mandibular dental-arch dimensions may affect the REI when using an MOA.

## 1. Introduction

Obstructive sleep apnea (OSA) is characterized by intermittent obstruction of the upper airway and recurrent episodes of apnea and hypopnea during sleep [1,2,3]. OSA is associated with obesity and abnormal maxillofacial morphology, such as micrognathia and mandibular recession [4]. Its symptoms include snoring, frequent urination at night, daytime sleepiness, and headache upon awakening [5,6]. Daytime sleepiness has been reported to decrease work efficiency and cause traffic accidents and work-related injuries [7]. Further, OSA has been reported to be a risk factor for cardiovascular diseases such as hypertension, myocardial infarction, and other systemic diseases such as diabetes mellitus and cerebrovascular diseases [8]. Thus, OSA becomes an issue for the patients themselves and society, and it is crucial to prevent these symptoms through early detection.

Although continuous positive airway pressure (CPAP) is the standard treatment for OSA, patient acceptance, tolerance, and treatment adherence are low, and the treatment process may be interrupted [9]. Alternatively, oral appliance (OA) therapy for OSA is simple, portable, and has recently been shown to be effective in some cases of severe OSA [10]. The OA for OSA is a device that secures the airway by advancing the mandible [11,12]. There are two types of OA: mono-block type, in which the upper and lower jaws are fixed, and bi-block type, in which the upper and lower jaws are separated [13]. In addition to OA for OSA, OA is also used for treating temporomandibular disorders (TMD) and sleep bruxism (SB) [14]. The most common type of OA for these treatments is the maxillary type OA (MOA), which acts by increasing the vertical intermaxillary distance without advancing the mandible. It has been reported that when patients with snoring or OSA wear an MOA during sleep, the snoring time increases, and the severity of OSA worsens [15].

Not every OSA patient has typical characteristics (male, middle-aged, obese, snoring, and sleepiness), and the subjective symptom such as quality of sleep, difficulty in breathing during sleep, and daytime fatigue is difficult to assess accurately. Therefore, OSA can be underdiagnosed and many potential patients may not be aware of themselves having OSA [4,16]. When these potential patients have TMD or SB, the easy use of an MOA as a treatment for these conditions may cause exacerbation of OSA without the patients noticing. In a study conducted by Gagnon et al. [15], about 40% of patients with snoring or OSA wearing an MOA during sleep showed a transition from snoring to OSA and worsening of the severity of OSA. Moreover, in a pilot study by Nikolopoulou et al. [17], raising the vertical dimension of OA with no mandibular advancement increased the apnea hypopnea index (AHI) at the individual level in 50% of their patients. We have also encountered cases in which such patients complained of breathlessness during sleep and had difficulty using an MOA, even though the possibility of OSA was relatively low. This finding suggests that the use of an MOA itself may adversely affect the respiratory variables during sleep. However, no studies have examined the respiratory variables during sleep with an MOA in patients without OSA. The effects of different MOA designs on respiratory status during sleep have also not been clarified. The purpose of this study was to investigate the effect of wearing or not wearing an MOA and of the difference in the width of the palatal coverage and the vertical intermaxillary thickness of an MOA on the respiratory variables during sleep in w/o-OSA or mild-OSA patients.

## 2. Materials and Methods

### 2.1. Study Design

This study was conducted to compare changes in respiratory variables on wearing an MOA during sleep. The MOAs used in this study were standard oral appliance (S-OA), palatal covering oral appliance (PC-OA), and vertically increasing oral appliance (VI-OA). Each OA was worn during sleep for three consecutive nights, with a seven-day wash-out period between each MOA experiment. This study was approved by the Ethics Committee of Tokyo Medical and Dental University (no. D2019-069) and conducted in accordance with the Declaration of Helsinki. The study was registered at UMIN-CTR (https://www.umin.ac.jp/ctr/ (accessed on 31 March 2020), UMIN000039637, Effect of palatal width of the maxillary oral appliance on respiratory status during sleep).

### 2.2. Participants

The candidates for this study were graduate students and staff of the Tokyo Medical and Dental University. The inclusion criteria were as follows: (1) body mass index (BMI) < 25.0, (2) Epworth Sleepiness Scale score < 11, (3) regular nighttime sleep, and (4) adequate dentition to support the OA. The exclusion criteria were as follows: (1) pain caused by TMDs; (2) severe sleep bruxism; (3) moderate to severe nasal disease (trouble breathing through the nose); (4) untreated, missing teeth excluding wisdom teeth; (5) ongoing medication for psychiatric disorders, including sleeping pills; (6) ongoing occlusal splint therapy. Before inclusion in the study, written informed consent was obtained from each subject.

### 2.3. Sleep Test

Before recruitment, a sleep study was performed using Watch-PAT (Itamar Medical, Caesarea, Israel), a portable sleep test device [18] consisting of a peripheral arterial tonometry probe, wrist-mounted device, and an oxygen saturation monitor, with a snoring monitor, to assess the state of OSA at baseline and follow-up. It has been reported that Watch-PAT is a simple, reliable device that significantly correlates with the gold standard polysomnography (PSG). Several studies have also reported that this device has excellent sensitivity and specificity as compared with in-laboratory polysomnography, and it is a highly sensitive portable device for estimating the treatment results of OSA by detecting overall AHI under various conditions of the patients [19,20,21,22]. In this study, the following values of the sleep test results were assessed: the respiratory event index (REI) and the lowest saturation of percutaneous oxygen (SpO_2_). The REI indicates the number of apnea and hypopnea events per hour of measurement. Apnea was defined as a 90% reduction in airflow for at least 10 s, and hypopnea was defined as ≥30% reduction in airflow for at least 10 s associated with ≥3% reduction in oxygen saturation [23]. OSA was defined as an REI ≥ 5 and classified as mild (REI 5.0–14.9), moderate (REI 15.0–29.9), and severe (REI ≥ 30) [24]. In this study, w/o-OSA participants (REI < 5) and mild-OSA participants (REI 5.0–14.9) were included. Subjects with REI ≥ 15.0 were excluded from the study.

### 2.4. Experimental Appliance Design and Measurement Schedule

Three types of MOAs (S-OA, PC-OA, and VI-OA) were fabricated for each participant. The standard measurements of each appliance design are shown in Figure 1. They were made of a 1.5 mm thick thermoplastic sheet (Sprint A, Square, YAMAHACHI DENTAL MFG., CO., Aichi, Japan). For each participant, a maxillary and mandibular dentition impression was taken with alginate impression material, and dental casts were fabricated. The dental casts were mounted on a semi-adjustable articulator in the maximum intercuspal position. A flat occlusal plane was created using self-curing resin (UNIFAST III; GC Corp., Tokyo, Japan) to allow group function and protrusive anterior guidance without corresponding interference at the chairside [25]. According to the measurement schedules, the OA design for each experiment was altered at the chairside (Figure 2). Each participant wore OAs in the order of PC-OA, S-OA, and VI-OA, consecutively for three nights, and a sleep test was performed on the last two nights. After the seven-day wash-out period of the PC-OA experiment, an S-OA was fabricated by shortening the PC-OA palatal flange for the S-OA experiment. The same procedure was applied to the VI-OA. For the VI-OA experiment, the occlusal surface of the S-OA was increased vertically by adding self-curing resin to achieve the required vertical thickness (5–8 mm). The thickness of the splint was measured between the upper and lower central incisors. 

### 2.5. Outcome Measures

As the primary outcome, we assessed REI with an MOA. As the secondary outcome, patient age, BMI, dental arch width and length, and sleep satisfaction while wearing the OAs were evaluated. In this study, based on the average REI value for the two nights during each MOA experiment, participants w/o-OSA with an REI ≥ 5.0 were categorized as the exacerbation group and those with an REI < 5.0 as the non-exacerbation group. Mild-OSA subjects with an REI ≥ 15.0 or REI ≥ baseline REI × 1.5 were categorized as the exacerbation group and those with an REI < 15.0 and REI < baseline REI × 1.5 as the non-exacerbation group. Then, the percentage of the exacerbation and non-exacerbation groups with the OA was evaluated in w/o-OSA and mild-OSA participants.

In this study, we analyzed the association between the maxillary and mandibular arch morphology and REI to investigate whether it affects the development and aggravation of OSA on wearing an MOA. The maxillary and mandibular dental-arch dimension was compared using dentition model analysis. The dental arch length (DAL), anterior dental arch width (A-DAW), and posterior dental arch width (P-DAW) were measured on each subject’s dental casts using a digital caliper and digital tread depth gauge to investigate the relationship between the maxillary and lower dentition, respectively (Figure 3). DAL represents the distance between the lowest point of the mesial incisal edge of the incisor and the highest point on the mesiobuccal cusp tip of the first molar, A-DAW represents the distance between the buccal cusp tips of the right and left first premolars, and P-DAW represents the distance between the mesiobuccal cusp tips of right and left first molars. The palatal depth (PD) was also measured for the maxillary cast. PD represents the vertical distance from the midpoint on the maxillary P-DAW line perpendicular to the palatal vault in the midline [26]. Moreover, the ratios of the maxillary to mandibular A-DAW and P-DAW (maxillary/mandibular) were also calculated [27].

The participants were asked to answer a questionnaire to indicate sleep quality, number of awakenings, and difficulty in breathing during sleep to assess sleep satisfaction while wearing each MOA. The questions (responses) were: “How was your sleep quality?” (0, bad; 1, somewhat bad; 2, somewhat good; and 3, good), “How many times did you wake up during sleep?” (0, not at all, 1, 1 time, 2, from 2 to 3 times, 3, more than 4 times), and “Did you experience difficulty in breathing while sleeping?” (0, no; 1, a little; 2, very).

### 2.6. Statistical Analysis

Continuous variables were described as mean ± standard deviation for variables with a normal distribution and as median (interquartile range) for variables with a non-normal distribution. Normality of distribution was assessed using the Shapiro–Wilk test. For the REI and lowest SpO_2_, the mean value of 2 nights while wearing an MOA was used as the representative value. The Welch t-test was used to compare age, BMI, REI, SpO_2_, and dentition model analysis between the exacerbation and non-exacerbation groups. For sleep satisfaction, the Mann–Whitney U test was used for comparison between the exacerbation and non-exacerbation groups. All statistical analyses were performed using the SPSS version 28.0 software (IBM, Tokyo, Japan), and the level of significance was set at 0.05.

## 3. Results

Thirty candidates (14 men and 16 women, mean age 30.4 years, range 26–35 years) who agreed to participate in this study underwent a sleep study to assess OSA. Five candidates were excluded based on an REI ≥ 15.0. Of the remaining 25 candidates, two dropped out during the experiment (one dropout due to loss of the OA, and one dropout due to difficulty sleeping when wearing the OA). In the end, the entire study protocol was completed with 23 participants (10 w/o-OSA participants and 13 mild-OSA participants).

### 3.1. Baseline Characteristics of the Participants

The participant characteristics at baseline are shown in Table 1. The mean age of all subjects was 31.1 ± 4.3 years, and their mean BMI was 21.9 ± 2.7 kg/m^2^. The mean REI of the participants was 5.9 ± 3.8 events/h. There was a significant difference in the REI, lowest SpO_2_, and A-DAW at baseline between the w/o-OSA participants and mild-OSA participants (*p* < 0.01). In contrast, there was no significant difference in age and BMI (*p* > 0.05). In the dentition model analysis, there were no significant differences in maxillary and mandibular measurements (maxillary DAL, P-DAW, PD, and mandibular DAL, A-DAW, and P-DAW) between the two groups (*p* > 0.05). Furthermore, there were no significant differences in DAL, P-DAW, DAL, A-DAW, P-DAW, and PD of the maxilla and mandible between the two groups (*p* > 0.05).

### 3.2. Comparison of Age, BMI, and Respiratory Variables in the Exacerbation and Non-Exacerbation Groups for Each MOA

The comparison of age, BMI, and respiratory variables wearing the three MOAs in the exacerbation and non-exacerbation groups is shown in Table 2. The exacerbation group among the w/o-OSA participants (*n* = 10) comprised 10.0% of the participants (*n* = 1) with S-OA, 40.0% (*n* = 4) with PC-OA, and 30.0% (*n* = 3) with VI-OA. The exacerbation group in mild-OSA (*n* = 13) consisted of 15.4% participants (*n* = 2) with S-OA, 23.1% (*n* = 3) with PC-OA, and 23.1% (*n* = 3) with VI-OA. On the one hand, in the w/o-OSA participants, there was a significant difference in the REI while wearing the three MOAs between the exacerbation and non-exacerbation groups (*p* < 0.05). On the other hand, there were no significant differences in age, BMI, and respiratory variables (baseline REI, baseline, and lowest SpO_2_ with the three MOAs) between the two groups (*p* > 0.05). In the mild-OSA group, there was a significant difference in the REI while wearing the PC-OA and VI-OA between the exacerbation and non-exacerbation groups (*p* < 0.05). However, there were no significant differences in age, BMI, baseline REI, and SpO_2_ between the exacerbated and non-exacerbated groups for each MOA group (*p* > 0.05).

### 3.3. The Relationship of Respiratory Variables between MOA Types and Dental Arch Morphology

The relationship of respiratory variables between MOA types and dental arch morphology is shown in Table 3. In w/o-OSA participants, while comparing the association between the presence of exacerbation of respiratory variables and dental arch morphology among OA types, the exacerbation group showed a significantly smaller mandibular P-DAW with S-OA than the non-exacerbation group (*p* < 0.05) w/o-OSA. In addition, the ratio of the maxillary to mandibular dental arch width (A-DAW) was significantly higher in the exacerbation group than in the non-exacerbation group for both PC-OA and VI-OA (*p* < 0.05). In mild-OSA participants, the exacerbation group while wearing VI-OA showed a significantly smaller PD than the non-exacerbation group (*p* < 0.05).

### 3.4. Sleep Satisfaction While Wearing the Three Types of MOAs

The sleep satisfaction while wearing three types of MOAs is shown in Table 4. In the subjective evaluation of sleep quality, arousal, and difficulty in breathing while wearing three types of MOAs, there was no significance between the exacerbation and non-exacerbation groups between different MOA types (*p* > 0.05).

## 4. Discussion

To the best of our knowledge, this study is the first to determine the effects of wearing an MOA on respiratory variables during sleep in w/o-OSA participants and mild-OSA participants. It has already been reported that MOA can worsen respiratory variables in patients with moderate or severe OSA. However, the effect of MOA use in w/o-OSA and mild-OSA patients has not been reported. It is also possible that patients with mild-OSA are unaware that they have OSA. Therefore, participants without OSA and with mild OSA were included in our study. In this study, a sleep study was conducted at the participants’ homes using Watch-PAT. PSG testing is desirable for more accurate measurement of respiratory status during sleep. However, it must be performed in a special laboratory, which is burdensome for patients. Repeated, non-disturbing measurements in the familiar sleep surroundings of a patient’s home to quantify the respiratory status was an advantage of our study methods. Although Tschopp et al. [28] found no evidence of first night effect and night-to-night variability in respiratory polygraphy while using Watch-PAT, Le Bon et al. [29] observed the first-night effect in their PSG study. We performed the sleep study for two nights for each MOA experiment and for all participants to strengthen the accuracy of our study.

In this study, exacerbation of the REI occurred in approximately 15–23% of the subjects in the mild-OSA group and from 10% to 40% of the subjects in the w/o-OSA group. Gagnon et al. [15] reported that in a PSG-based study, 40% of patients with snoring or sleep apnea showed an increase in AHI on using an MOA. The MOA used in that study was approximately 1.5 mm thick at the molar level and covered all teeth and the anterior midpalatal area which could be considered similar to PC-OA in our study with a slight difference in vertical thickness (2–3 mm). Nikolopolou et al. [30] also reported that in a randomized study of patients with OSA, all patients had an increased AHI on using an MOA which had no palatal coverage and a bite rise of about 1 mm at molar level. This MOA is considered to be similar to the S-OA we used. Those findings indicated, once again, that wearing MOAs may worsen the respiratory variables of mild-OSA patients during sleep. In addition, 10–40% of w/o-OSA subjects in this study had exacerbation of the REI after wearing an MOA. We cannot compare the results of our study with those of the previous studies because those studies have not been conducted under similar conditions. However, we believe that the results confirm our clinical realization that some patients experience breathlessness during sleep using MOAs.

The position of the mandible and tongue may be related to worsening of the respiratory status during sleep due to the wearing of an MOA. In w/o-OSA sleepers, the airway collapses during sleep, particularly in the supine position, which in turn causes opening of the mandible and posterior displacement of the tongue [31,32]. Similar findings have been reported in OSA patients with more giant tongues [33,34,35]. Moreover, anatomically, the pharyngeal airway, similar to a collapsible tube, is enclosed by the tongue, mandible, and cervical vertebrae. When the vertical distance between the upper and lower jaw increases, the posteriorly retruded mandible positions the tongue in the same direction. The backward displacement of the tongue reduces the pharyngeal airway size, increasing airway collapsibility at both the velopharynx and oropharynx levels [36]. These findings suggest that the increased vertical distance between the maxilla and mandible due to the MOA affect the exacerbation of the respiratory status during sleep. A more significant proportion of subjects with VI-OA had a worse REI score than those with S-OA, both in the w/o-OSA and mild-OSA groups. The VI-OA had the same palatal coverage width as the S-OA but was set to increase the vertical distance between the maxillary and mandible. It has been reported that increasing the vertical dimension with occlusal orthotics reduced the AHI in non-supine and supine positions in the orthotic group [37]. The greater vertical distance between the maxilla and mandible may have caused the mandible to retract further, obstructing the airway with the tongue and causing further exacerbation of the REI.

In this study, a larger proportion of patients with PC-OA had a worse REI score than those with S-OA, both in the w/o-OSA and mild-OSA groups. The PW-OA had the same vertical distance between the maxilla and mandible as the S-OA but was set to increase the width of the palatal mucosa coverage. This may have resulted in a narrower space for the tongue relative to the palate as compared with the S-OA. The ideal resting tongue posture or maintenance of the anterior oral seal was acquired by naturally breathing through the nose and posturing the tongue correctly against the palate while growing up if there was nothing to interfere with this natural tongue position, such as chronic nasal congestion, retrognathia, and iatrogenic posterior mandibular displacement [38]. In this situation, the resting tongue naturally guides the maxillary arch into a parabolic/U-shape associated with obligate nasal breathers and a narrower V-shape associated with complete mouth breathers. In individuals with the “proper” tongue to palate posture, slight changes can disrupt the maintenance of this delicate posterior oral seal during sleep and result in the tongue falling back into the throat during sleep, especially when supine. However, in individuals who have already adapted (complete oral breathers and sometimes also oronasal breathers) to such conditions via other mechanisms, their tongues are presumably not resting against the palate. Therefore, in this study, the intrusion of the PC-OA into the palatal space may have caused a marked difference in the change in the REI in the exacerbation group.

In this study, while comparing the upper dental arch measurements between exacerbation and non-exacerbation groups in w/o-OSA participants and mild-OSA participants, A-DAW and P-DAW were greater in the exacerbation group than in the non-exacerbation group. Maxillofacial abnormalities, such as micrognathia and retrognathia, are widely known risk factors for OSA [39]. Moreover, since a constricted maxillary arch could also decrease the oropharyngeal volume in the Donder’s space for the tongue, a narrow maxillary dental arch is a risk factor for OSA [27]. Some studies have suggested that a narrow upper dental arch was correlated with sleep disturbance in OSA patients [40,41,42]. However, Maeda et al. [27] reported that the upper dental arch was wider when OSA patients were more obese and the lower dental arch was more expanded, and the upper dental arch became narrower when the mandible was in a more backward position relative to the maxilla. DAL was significantly shorter in the exacerbation group than in the non-exacerbation group. In a study of Malaysian patients, Banabilh et al. [43] reported that OSA patients had longer maxillary and mandibular dental arches than controls. These reports are not consistent with the results of the present study. Since the BMI of the subjects in this study was within the normal range, it can be inferred that the exacerbation of REI with PC-OA and VI-OA is not due to the dental morphology, but the effect of MOA wear itself.

Although several publications have supported the fact that REI was worsened with MOA during sleep, it is worth notable that there was an improvement in REI in the non-exacerbation group of mild-OSA subjects (SOA—84.6%, PC-OA—76.9%, and VI-OA—76.9%) in this study. In a randomized trial, Kuwashima et al. [37] reported a decrease in AHI with MOA for mild OSA. Anitua et al. also showed a decrease in AHI with the use of upper and lower integrated OA without mandibular advancement, and concluded that elevation of the occlusal height diameter alone could maintain upper airway patency and stabilize pharyngeal tissue.

This study has several limitations. The assignment of the three MOAs was not randomized in this study. Therefore, it is possible that there was an order effect of MOA on the study results although there was a wash-out period of seven days between each MOA experiment. The participants were asked to answer a questionnaire to indicate sleep quality, number of awakenings, and extent of difficulty in breathing during sleep to assess sleep satisfaction while wearing each MOA. However, there was no significant difference in sleep satisfaction among both w/o-OSA and mild-OSA participants for any MOA. This result may have been influenced by the small number of participants with worse REI during sleep and the small overall sample size. The same may be true for other outcomes. In this study, it was suggested that wearing an MOA may exacerbate the respiratory variables during sleep not only in mild-OSA participants but also in w/o-OSA participants. Furthermore, the increased palatal coverage area of the MOA and increased vertical distance between the maxilla and mandible also increased the risk of exacerbating the rate of breathing during sleep. However, no obvious characteristics were found in the dental morphology of the upper and lower jaws between those whose breathing conditions worsened and those whose breathing conditions did not. Regarding the size of the dentition, no sex difference was also found in the data. The three-dimensional positioning of the mandible in relation to the maxilla and the anatomy of the nasopharynx may also be involved, and these factors should be examined in the future. Therefore, in a future study, the three-dimensional positioning of the mandible in relation to the maxilla and the anatomy of the nasopharynx should be evaluated by cephalometric and CT scan parameters. Moreover, although the possible impact of PC-OA and VI-OA design on respiratory variables during sleep was explained by the relationship between mouth breathing and the anterior oral seal in this study, validation of the impact of disruption of the tongue’s space and anterior oral seal on the respiratory variables should be evaluated by including nasal and oral breathers in the sample in the future. This study did not consider changes in respiratory variables in the different sleep postures. To prove the disruption of the normal tongue position by the MOAs, the difference in change in the REI during supine and non-supine should be considered in the future. 

## 5. Conclusions

In conclusion, the study findings confirmed that, in w/o-OSA and mild-OSA participants, wearing an MOA may increase the respiratory variables during sleep and that PC-OA and VI-OA may increase respiratory variables more than S-OA. The maxillary and mandibular dental-arch dimensions may affect the respiratory variables of an MOA. Therefore, when an MOA is indicated for patients with SB or TMD, it is necessary to consider the possibility of worsening of the respiratory variables during sleep and the extent of palatal coverage of the MOA, and the vertical thickness of the OA. The importance of confirming the presence of OSA in advance was also indicated. In future studies, three-dimensional assessment of dental morphology with MOAs during different sleep positions should be carried out with a larger sample size to better understand the role of the tongue and soft palate in the exacerbation of the REI by OA application.

## Figures and Tables

**Figure 1 ijerph-19-06714-f001:**
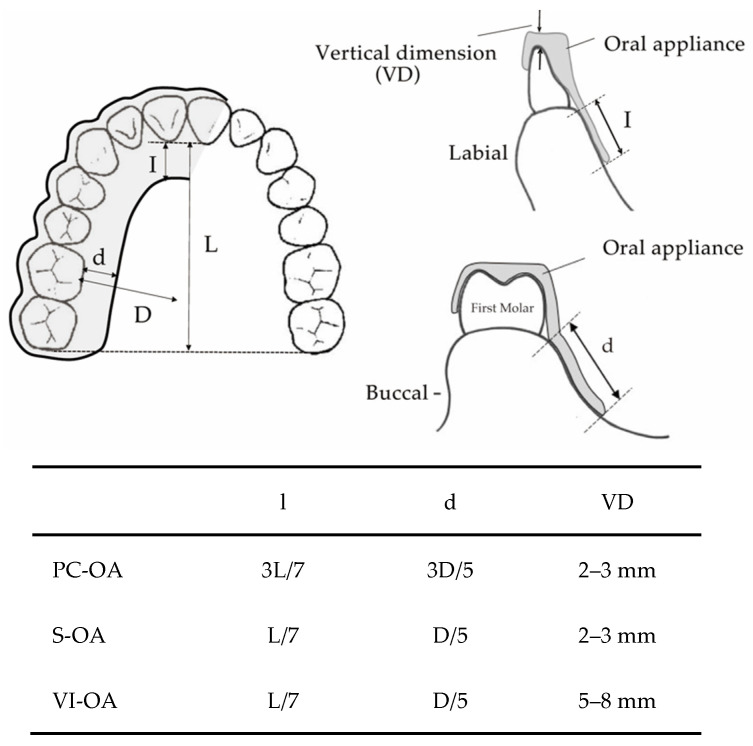
The standard measurements of each OA design.

**Figure 2 ijerph-19-06714-f002:**
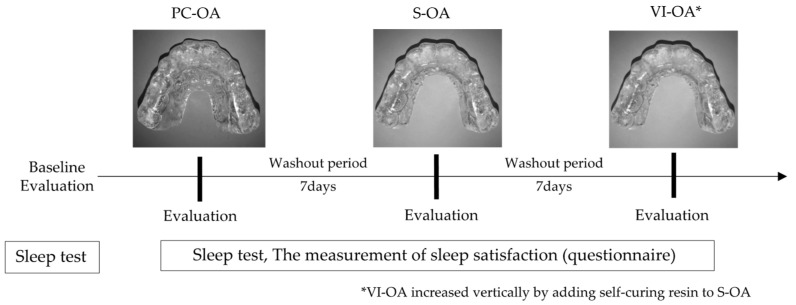
Measurement schedules of each OA design.

**Figure 3 ijerph-19-06714-f003:**
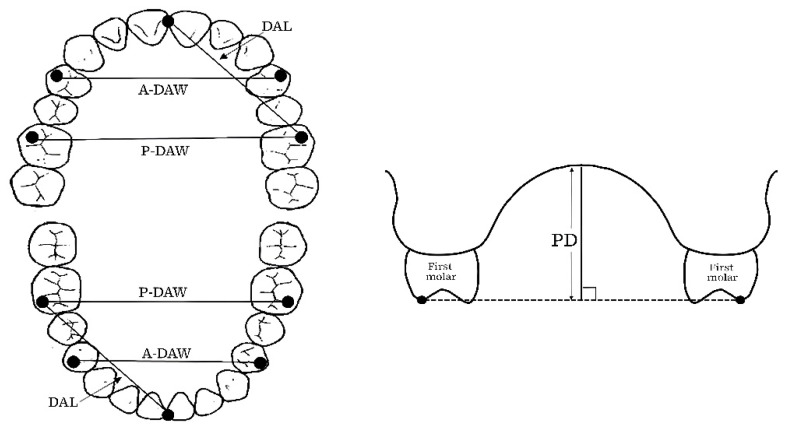
Measurement parameters of the maxillary and mandible on plaster casts.

**Table 1 ijerph-19-06714-t001:** Participant characteristics of the w/o-OSA and mild-OSA at baseline.

	Total	w/o-OSA Participants (*n* = 10)	Mild-OSA Participants (*n* = 13)	*p*-Value ^a^
Age (years)	31.1 (4.3)	29.9 (4.1)	32.1 (4.3)	0.24
BMI (kg/m^2^)	21.9 (2.7)	22.4 (2.7)	21.6 (2.7)	0.45
REI (events/h)	5.9 (3.8)	2.3 (1.1)	8.7 (2.7)	<0.001
Lowest SpO_2_ (%)	91.7 (2.4)	93.2 (1.3)	90.5 (2.4)	0.003
Maxillary DAL (mm)	44.2 (2.9)	45.2 (3.8)	43.5 (1.9)	0.21
A-DAW (mm)	43.0 (3.4)	44.9 (3.1)	41.6 (2.8)	0.017
P-DAW (mm)	55.6 (3.0)	56.5 (2.7)	54.8 (3.1)	0.17
PD (mm)	20.5 (1.8)	20.4 (1.6)	20.6 (2.0)	0.83
Mandibular DAL (mm)	31.3 (2.8)	32.4 (3.0)	30.5 (2.3)	0.12
A-DAW (mm)	34.0 (3.5)	35.4 (3.0)	32.9 (3.7)	0.087
P-DAW (mm)	47.7 (3.1)	48.2 (2.2)	47.0 (3.6)	0.23
Maxillary/Mandibular: A-DAW	1.27 (0.09)	1.27 (0.09)	1.27 (0.09)	0.93
Maxillary/Mandibular: P-DAW	1.17 (0.04)	1.17 (0.04)	1.17 (0.04)	0.85

Data expressed as mean ± standard deviation (SD). ^a^ Comparison between w/o-OSA and mild-OSA participant groups. BMI, body mass index; REI, respiratory event index; Lowest SpO_2_, lowest saturation of percutaneous oxygen; DAL, dental arch length; A-DAW, anterior dental arch width; P-DAW, posterior dental arch width; PD, palatal depth.

**Table 2 ijerph-19-06714-t002:** Comparison of age, BMI, and respiratory variables in exacerbation and non-exacerbation groups by wearing OA.

		Number (%)	Age (years)	BMI (kg/m^2^)	Baseline		With OA	
					REI (events/h)	Lowest SpO_2_ (%)	REI (events/h)	Lowest SpO_2_ (%)
w/o-OSA participants (*n* = 10)								
S-OA	Exacerbation	1 (10%)	35.0 (-)	22.3 (-)	4.2 (-)	92.0 (-)	6.8 (-)	91.5 (-)
	Non-exacerbation	9 (90%)	29.3 (4.0)	22.5 (2.9)	2.1 (1.0)	93.3 (1.3)	2.7 (1.0)	91.8 (2.2)
	*p*-Value		0.21	0.96	0.088	0.38	0.005	0.89
PC-OA	Exacerbation	4 (40%)	29.3 (4.8)	21.8 (2.7)	2.8 (1.2)	93.88 (1.3)	7.7 (0.85)	93.1 (2.2)
	Non-exacerbation	6 (60%)	30.3 (4.1)	22.9 (2.9)	2.0 (1.1)	92.75 (1.3)	2.5 (1.0)	93.3 (1.6)
	*p*-Value		0.72	0.56	0.29	0.22	<0.001	0.93
VI-OA	Exacerbation	3 (30%)	31.0 (4.0)	20.6 (1.8)	2.8 (1.5)	93.67 (1.5)	8.8 (2.4)	89.0 (4.0)
	Non-exacerbation	7 (70%)	29.4 (4.4)	23.2 (2.8)	2.1 (1.1)	93.00 (1.3)	2.6 (0.66)	93.6 (1.3)
	*p*-Value		0.61	0.13	0.53	0.55	0.042	0.18
Mild-OSA participants (*n* = 13)								
S-OA	Exacerbation	2 (15.4%)	29.5 (5.0)	19.1 (1.4)	7.3 (0.92)	92.5 (2.1)	16.9 (5.7)	92.5 (2.1)
	Non-exacerbation	11 (84.6%)	32.6 (4.3)	22.0 (2.6)	8.9 (2.8)	90.09 (2.4)	5.8 (3.7)	92.0 (2.1)
	*p*-Value		0.54	0.12	0.18	0.32	0.20	0.80
PC-OA	Exacerbation	3 (23.1%)	36.3 (4.2)	22.1 (3.6)	7.0 (2.2)	92.0 (1.0)	16.0 (1.1)	85.5 (6.9)
	Non-exacerbation	10 (76.9%)	30.8 (3.6)	21.4 (2.6)	9.1 (2.7)	90.0 (2.6)	4.7 (2.8)	92.5 (2.3)
	*p*-Value		0.13	0.78	0.24	0.074	<0.001	0.22
VI-OA	Exacerbation	3 (23.1%)	30.3 (4.0)	22.4 (2.0)	10.8 (3.5)	90.8 (2.8)	15.1 (2.0)	90.0 (4.6)
	Non-exacerbation	10 (76.9%)	32.6 (4.5)	21.3 (2.9)	8.0 (2.2)	90.4 (2.5)	5.3 (3.3)	91.9 (2.6)
	*p*-Value		0.46	0.49	0.30	0.81	0.001	0.56

Data expressed as mean (standard deviation, SD). S-OA, standard-OA; PC-OA, palatal covering-OA; VI-OA, vertical increasing-OA; BMI, body mass index; REI, respiratory event index; Lowest SpO_2_, lowest saturation of percutaneous oxygen.

**Table 3 ijerph-19-06714-t003:** The relationship between changes in respiratory status and dentition morphology in types of OA.

		Maxillary Dental Arch (mm)				Mandibular Dental Arch (mm)			Maxillary/Mandibular	
		DAL	A-DAW	P-DAW	PD	DAL	A-DAW	P-DAW	A-DAW	P-DAW
w/o-OSA participants (*n* = 10)										
S-OA	Exacerbation	40.0 (-)	45.8 (-)	60.4 (-)	20.6 (-)	32.0 (-)	38.9 (-)	52.6 (-)	1.18 (-)	1.15 (-)
	Non-exacerbation	45.8 (3.5)	44.8 (3.3)	56.1 (2.5)	20.4 (1.7)	32.5 (3.2)	35.0 (2.9)	48.1 (1.7)	1.28 (0.09)	1.17 (0.04)
	*p*-Value	0.16	0.77	0.14	0.95	0.90	0.23	0.040	0.29	0.67
PC-OA	Exacerbation	44.0 (3.3)	44.1 (1.3)	56.3 (3.2)	20.4 (0.1)	31.8 (0.9)	37.4 (1.3)	49.4 (2.2)	1.18 (0.04)	1.14 (0.05)
	Non-exacerbation	46.0 (4.1)	45.4 (4.0)	56.7 (2.6)	20.5 (2.2)	32.9 (3.9)	34.1(3.1)	47.8 (2.2)	1.33 (0.04)	1.18 (0.02)
	*p*-Value	0.42	0.49	0.88	0.98	0.54	0.051	0.32	0.001	0.15
VI-OA	Exacerbation	42.7 (2.3)	44.6 (1.1)	56.5 (3.9)	20.4 (0.2)	32.0 (1.0)	37.5 (1.6)	49.6 (2.6)	1.19 (0.03)	1.17 (0.06)
	Non-exacerbation	46.3 (4.0)	45.0 (3.8)	56.5 (2.4)	20.5 (2.0)	32.6 (3.7)	34.6 (3.1)	48.0 (2.0)	1.30 (0.08)	1.18 (0.02)
	*p*-Value	0.11	0.82	1.0	0.97	0.71	0.093	0.41	0.015	0.37
Mild-OSA participants (*n* = 13)										
S-OA	Exacerbation	44.5 (0.7)	41.3 (3.5)	53.5 (5.1)	21.2 (1.6)	29.3 (3.8)	34.3 (8.2)	47.2 (7.5)	1.23 (0.19)	1.14 (0.07)
	Non-exacerbation	43.3 (2.0)	41.6 (2.9)	55.1 (2.9)	20.5 (2.2)	30.7 (2.2)	32.7 (3.0)	47.0 (3.2)	1.28 (0.08)	1.17 (0.04)
	*p*-Value	0.18	0.93	0.73	0.66	0.69	0.83	0.97	0.77	0.61
PC-OA	Exacerbation	43.3 (2.1)	43.2 (4.0)	53.3 (3.0)	19.6 (0.6)	30.0 (2.0)	34.1 (4.8)	45.9 (4.2)	1.27 (0.08)	1.16 (0.06)
	Non-exacerbation	43.5 (2.0)	41.1 (2.4)	55.3 (3.1)	20.9 (2.2)	30.7 (2.5)	32.6 (3.5)	47.3 (3.6)	1.27 (0.10)	1.17 (0.04)
	*p*-Value	0.91	0.46	0.39	0.10	0.65	0.66	0.91	0.90	0.84
VI-OA	Exacerbation	44.3 (2.5)	42.9 (1.8)	56.9 (2.0)	22.2 (0.3)	29.2 (3.3)	35.1 (4.4)	49.0 (3.7)	1.23 (0.12)	1.17 (0.07)
	Non-exacerbation	43.2 (1.8)	41.2 (3.0)	54.2 (3.2)	20.1 (2.1)	30.9 (2.0)	32.3 (3.4)	46.4 (3.6)	1.28 (0.08)	1.17 (0.04)
	*p*-Value	0.53	0.29	0.14	0.012	0.48	0.38	0.36	0.53	0.94

Data expressed as mean (standard deviation, SD). S-OA, standard-OA; PC-OA, palatal covering-OA; VI-OA, vertical increasing-OA; DAL, dental arch length; A-DAW, anterior dental arch width; P-DAW, posterior dental arch width; PD, palatal depth.

**Table 4 ijerph-19-06714-t004:** Sleep satisfaction while wearing the three types of MOAs.

		Sleep Quality ^a^	Arousal ^b^	Difficulty in Breathing ^c^
w/o-OSA participants (*n* = 10)				
S-OA	Exacerbation	2.5 (2.5, 2.5)	2.0 (2.0, 2.0)	0.0 (0.0, 0.0)
	Non-exacerbation	2.0 (1.3, 3.0)	0.5 (0.0, 1.8)	0.0 (0.0, 0.5)
	*p*-Value	0.80	0.40	0.80
PC-OA	Exacerbation	3.0 (3.0, 3.0)	0.0 (0.0, 0.0)	0.0 (0.0, 0.0)
	Non-exacerbation	2.0 (1.5, 3.0)	1.0 (0.0, 2.0)	0.0 (0.0, 1.0)
	*p*-Value	0.40	0.40	0.60
VI-OA	Exacerbation	3.0 (3.0, 3.0)	0.5 (0.5, 0.5)	0.0 (0.0, 0.0)
	Non-exacerbation	3.0 (1.5, 3.0)	1.0 (0.5, 1.5)	0.0 (0.0, 0.3)
	*p*-Value	0.80	0.60	0.80
Mild-OSA participants (*n* = 13)				
S-OA	Exacerbation	1.8 (1.4, 2.1)	1.8 (1.6, 1.9)	0.5 (0.3, 0.8)
	Non-exacerbation	2.0 (1.5, 2.4)	1.0 (0.6, 1.9)	0.0 (0.0, 0.4)
	*p*-Value	0.77	0.31	0.64
PC-OA	Exacerbation	1.5 (0.4, 1.5)	1.5 (1.5, 1.5)	0.0 (0.5, 1.0)
	Non-exacerbation	1.5 (1.0, 1.9)	1.5 (1.0, 1.5)	0.0 (0.0, 0.9)
	*p*-Value	0.57	0.81	1.0
VI-OA	Exacerbation	2.0 (2.1, 2.3)	1.0 (0.6, 1.0)	0.0 (0.0, 0.0)
	Non-exacerbation	1.8 (1.1, 2.0)	1.3 (1.0, 2.0)	0.5 (0.0, 1.0)
	*p*-Value	0.11	0.29	0.16

Data expressed as median and interquartile range (IQR). ^a^ 0, bad; 1, somewhat bad; 2, somewhat good; 3, good. ^b^ 0, not at all; 1, 1 time; 2, from 2 to 3 times; 3, more than 4 times. ^c^ 0, no; 1, a little; 2, very. S-OA, standard-OA; PC-OA, palatal covering-OA; VI-OA, vertical increasing-OA.

## Data Availability

The data presented in this study are available on request from the corresponding author.

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
