# Peer review of "Effect of Different Maxillary Oral Appliance Designs on Respiratory Variables during Sleep"

_ijerph, 2022, doi:10.3390/ijerph19116714_

Round 1

Reviewer 1 Report

What does "normal participans" mean? It would be better to write “without OSA”.

How much was the VDO increase in VI-OA? The thickness of the splint and other value increases in different types of occlusion, moreover, it is necessary to specify where the thickness of the splint was measured and how much it increased the occlusion.

Please change the diagram of the buccal cusps so that the palatine cusps are in contact with the plane and not the articulating cusps (Fig. 3).

  Too small group - I think it's worth writing it in the title.

Author Response

Point 1: What does "normal participants" mean? It would be better to write “without OSA”.

Response 1: Thanks for pointing that out, we rewrote “normal participants” to “participants without OSA (w/o-OSA)”.

Point 2: How much was the VDO increase in VI-OA? The thickness of the splint and other value increases in different types of occlusion, moreover, it is necessary to specify where the thickness of the splint was measured and how much it increased the occlusion.

Response 2: The VDO increased in VI-OA is 5-8 mm. The thickness of the splint was measured between upper and lower central incisors. We have included an additional explanation in the text (line: 138-141).

Point 3: Please change the diagram of the buccal cusps so that the palatine cusps are in contact with the plane and not the articulating cusps (Fig. 3).

Response 3: We corrected the Figure 3 as you indicated.

Point 4: Too small group - I think it's worth writing it in the title. Should I correct the title according to this comment?

Response 4: We would like to leave the title as it is. Instead, we have included the sample size as a limitation of the study in the discussion.

Reviewer 2 Report

Intresting article, draws attention to problemsthat may occur during use MOA. The main limitations were mentioned by the authors: small sample, no sleep position taken into account, but also no gender division of the patients and no questionnaire was conducted before the measurements were taken. What are the clinical implications according to the authors?  

Some parts of publication need to be improved.

1.) "... The first choice for OSA treatment is continuous positive airway pressure therapy, but this treatment has low treatment adherence [9].

2.)"... Becouse many patients with OSA have no subjective symptoms, it is thought that there are many undiagnosed potential patients [16]. When these potential patients have TMD or SB, the easy use of an MOA as a treatment for these conditions may cause exacerbation of OSA without the patients patients noticing [15].On the other hand, we have also encountered cases in which such patients complained ...

Methods.

- it would be easier to compare groups with the same number of patients:

- when comparing the width of the arches, it worth taking into account the sex of the respondens.

"The participans were asked to answer the questionnaite to indicate sleep quality, number of awakenings, and difficulty in breathing during sleep satisfaction while wearing each MOA" - Why there is no survey measurements with MOA?

- In chapter 2.4 it is enough to write information about the measurement time once fot all three oral appliances (... was worn for three nights consecutively, and a sleep test was performed on the last two nights ..."

Results:

- Table 2 All (n=10) - ? only for the normal group and where for the mild OSA (n=13)?

- It is recommended to rewritr the sentence to make it more accessible;

230-233 " While comparing the association between respiratory variables and dental arch ..... than the non-exacerbation group (p<0.05)."

- Figure 3; there is no need to describe the measurement parameters twice; in the text and below the figure.

- Table 4. the values should be described under the table.

- line 262: PSG - the abbreviation should be extended.

- 272-276: It is recommended to rewrite the sentence; "Although we cannot compare the result for our study ...... with the use of MOAs.

Author Response

Point 1:  Some parts of publication need to be improved.

1.)"... The first choice for OSA treatment is continuous positive airway pressure therapy, but this treatment has low treatment adherence [9].

2.)"... Because many patients with OSA have no subjective symptoms, it is thought that there are many undiagnosed potential patients [16]. When these potential patients have TMD or SB, the easy use of an MOA as a treatment for these conditions may cause exacerbation of OSA without the patients noticing [15]. On the other hand, we have also encountered cases in which such patients complained ...

Response 1: We changed the manuscript as follows.

 “Although continuous positive airway pressure (CPAP) is the standard treatment for OSA, patient acceptance, tolerance, and treatment adherence are low and the treatment process may be interrupted [9].”

“Not every OSA patient has typical characteristics (male, middle-aged, obese, snoring, and sleepiness), and the subjective symptom such as quality of sleep, difficulty in breathing during sleep, and daytime fatigue is difficult to assess accurately. Therefore, OSA can be under-diagnosed and many potential patients may not be aware of themselves having OSA [16,17] .”

Point 2:  Methods.

- it would be easier to compare groups with the same number of patients:

- when comparing the width of the arches, it worth taking into account the sex of the respondens.

Response 2: We added to the study limitations that the number of people in the w/o-OSA and mild-OSA groups differed. Regarding the size of the dentition, no sex difference was found in the data.

Point 3: "The participants were asked to answer the questionnaires to indicate sleep quality, number of awakenings, and difficulty in breathing during sleep satisfaction while wearing each MOA" - Why there is no survey measurements with MOA?

Response 3: We guessed that "with MOA" was a mistake for "without MOA". The purpose of this study was to confirm the quality of sleep with MOA compared to without MOA. However, as you pointed out, it would have been better to conduct the study when the patients were not wearing MOA. Thank you for your suggestion.

Point 4: - In chapter 2.4 it is enough to write information about the measurement time once fot all three oral appliances (... was worn for three nights consecutively, and a sleep test was performed on the last two nights ..."

Response 4: We changed the manuscript as follows (line: 134-140).

“Each participant wore OAs in the order of PW-OA, S-OA, and VI-OA, consecutively for three nights, and a sleep test was performed on the last two nights. After the seven-day wash-out period of the PW-OA experiment, an S-OA was fabricated by shortening the PW-MOA palatal flange for the S-OA experiment the same procedure was applied for the VI-OA. For the VI-OA experiment, the occlusal surface of the S-OA was increased vertically by adding self-curing resin to achieve the required vertical thickness (5-8 mm).”

Point 5: Results:

- Table 2 All (n=10) - ? only for the normal group and where for the mild OSA (n=13)?

Response 5: The "n=10" in the table has been deleted. Instead, the number of participants has been added after each group name.

Point 6: - It is recommended to rewrite the sentence to make it more accessible;

230-233 " While comparing the association between respiratory variables and dental arch ..... than the non-exacerbation group (p<0.05)."

Response 6: We changed the manuscript as follows (line:251-254).

“In w/o-OSA participants, while comparing the association between respiratory variables and dental arch morphology among OA types, the exacerbation group showed a significantly smaller mandibular P-DAW with S-OA than the non-exacerbation group (p<0.05).”

Point 7: - Figure 3; there is no need to describe the measurement parameters twice; in the text and below the figure.

Response 7: We deleted the parameters under Figure 3.

Point 8:  - Table 4. the values should be described under the table.

Response 8: We described the values under Table 4.

Point 9: - line 262: PSG - the abbreviation should be extended.

Response 9: We included the original wording as you indicated.

Point 10: - 272-276: It is recommended to rewrite the sentence; "Although we cannot compare the result for our study ...... with the use of MOAs.

Response 10: We changed the manuscript as follows (line:318-322).

“We cannot compare the results of our study with those of the previous studies because these studies have not been conducted under similar conditions. However, we believe that the results confirm our clinical realization that some patients experience breathlessness during sleep using MOAs.”

Reviewer 3 Report

This is a clinically very important research report because it shows that the application and form of oral appliance may exacerbate respiration during sleep and that dental morphology is a risk factor for its adverse effects in both healthy subjects and mild OSA patients. There are a few areas that need to be corrected before acceptance, so please use the following as a guide to revise your paper

  1. Palatal widening-OA sounds like an orthodontic device, so Palatal covering-OA would be more appropriate. Please consider.

  1. Gagnon's OA is "It was approximately 1.5 mm thick at the molar level and covered all teeth and the anterior mid palatal area. It is a so-called Michigan type splint, which I believe is the same form as the author's PW-OA. On the other hand, Nikolopoulou's OA is a stabilization splint, the same as the author's S-OA. They also report only a slight decrease in AHI. Please revise the third paragraph of Introduction and the second paragraph of Discussion to note the different forms of their OAs and the different results by their OAs.

  1. Please explain in Materials and Methods or Discussion why moderate and severe OSA patients were excluded (2.3. Sleep Test).

  1. Did patients who worsened with S-OA also worsen with other OAs? If you can show the results of this overlap, it would provide support to suggest further deterioration due to palatal coverage or occlusal elevation. Please consider.

  1. From Table 2, there seem to be many cases in which the REI improved. Please discuss them as well.

Author Response

Point 1: Palatal widening-OA sounds like an orthodontic device, so Palatal covering-OA would be more appropriate. Please consider.

Response 1: We agree with the reviewer. We rewrote the Palatal coverage-OA in the manuscript.  

Point 2: Gagnon's OA is "It was approximately 1.5 mm thick at the molar level and covered all teeth and the anterior mid palatal area. It is a so-called Michigan type splint, which I believe is the same form as the author's PW-OA. On the other hand, Nikolopoulou's OA is a stabilization splint, the same as the author's S-OA. They also report only a slight decrease in AHI. Please revise the third paragraph of Introduction and the second paragraph of Discussion to note the different forms of their OAs and the different results by their OAs.

Response 2: We agree with the reviewer. We changed the manuscript.

Point 3: Please explain in Materials and Methods or Discussion why moderate and severe OSA patients were excluded (2.3. Sleep Test).

Response 3: We changed the manuscript as follows (line:293-297).

“It has already been reported that MOA can worsen respiratory variables in patients with moderate or severe OSA. However, the effect of MOA use in w/o-OSA and mild-OSA patients has not been reported. It is also possible that patients with mild-OSA are unaware that they have OSA. Therefore, participants without OSA and with mild OSA were included in our study.”

Point 4: Did patients who worsened with S-OA also worsen with other OAs? If you can show the results of this overlap, it would provide support to suggest further deterioration due to palatal coverage or occlusal elevation. Please consider.

Response 4: As to whether participants with worsening REI in S-OA showed a trend toward worsening in other types of OA. No clear pattern existed. This may be due to the sample size.

Point 5: From Table 2, there seem to be many cases in which the REI improved. Please discuss them as well.

Response 5: We added explanatory text to the discussion as follows (line:378-385 ).

Reviewer 4 Report

Congratulations for improving this manuscript. 

Minor comments- 

C1. What are the characteristics of patients who were excluded? 7 of 30.

C2. Please cite Watch-PAT correlation with in lab PSG (gold standard) in methods.

C3. Prior studies with maxillomandibular advancement surgery have reported a greater success than mandibular advancement devices by itself (Garreau et al. PMID: 24923216). Can the authors comment on any potential comparison between maxillomandibular advancement surgery vs maxillary oral appliances?

C4. Would not call study sample large as noted in line 257 in Discussion.

Author Response

Point 1: What are the characteristics of patients who were excluded? 7 of 30.

Response 1: Among seven participants, two were excluded during study. One male and female (mild-obstructive apnea, mean age(standard deviation): 29.5 (3.53), and BMI is lower than 25). The five participants were excluded before the study by exclusion criteria.

Point 2: Please cite Watch-PAT correlation with in lab PSG (gold standard) in methods.

Response 2: We added the manuscript as follows (line:115-110).

“It is reported that Watch-PAT is a simple, reliable device that significantly correlated with the gold standard polysomnography (PSG). Several studies had also been reported that this device has excellent sensitivity and specificity compared with in-laboratory polysomnography, and it is a highly sensitive portable device for estimating the treatment results of OSA by detecting overall AHI under various conditions of the patients [20-23].”

Point 3:  Prior studies with maxillomandibular advancement surgery have reported a greater success than mandibular advancement devices by themselves (Garreau et al. PMID: 24923216). Can the authors comment on any potential comparison between maxillomandibular advancement surgery vs maxillary oral appliances?

Response 3: In my opinion, if we compare maxillomandibular advancement surgery (MMA) and maxillary oral appliances themselves, the treatment outcome would be more favorable to MMA than MOA because wearing MOA alone without protruding mandible would have aggravated the respiratory condition during sleep as we found in our study and other previous studies. However, the application of MOA after MMA surgery would be beneficial for the SB and TMD patients with OSA and concomitant dentofacial deformity. However, because this study will focus on the harms of MOA placement for non-apneic and mildly apneic patients, we will not discuss the effectiveness of treatment of apnea in this manuscript because it is not consistent with the focus of this study.

Point 4: Would not call study sample large as noted in line 257 in Discussion.

Response 4: Thank you for pointing this out. We have corrected the wording.